# Cardiovascular biomarkers as predictors of adverse outcomes in chronic Chagas cardiomyopathy

Luis E. Echeverría[1,2]*, Lyda Z. Rojas[3], Sergio Alejandro Gómez-Ochoa[2], Oscar L. Rueda-Ochoa[4], Cristian David Sosa-Vesga[5], Taulant Muka[6], James L. Januzzi[7], Rachel Marcus[8], Carlos A. Morillo[9,10]

1 Heart Failure and Cardiac Transplant Division, Fundación Cardiovascular de Colombia, Floridablanca, Colombia, 2 Public Health and Epidemiological Studies Group, Cardiovascular Foundation of Colombia, Floridablanca, Colombia, 3 Research Group and Development of Nursing Knowledge (GIDCEN-FCV), Research Center, Cardiovascular Foundation of Colombia, Floridablanca, Santander, Colombia, 4 Electrocardiography Research Group, Medicine School, Universidad Industrial de Santander, Bucaramanga, Colombia, 5 Universidad Industrial de Santander, Bucaramanga, Santander, Colombia, 6 Institute of Social and Preventive Medicine (ISPM), University of Bern, Bern, Switzerland, 7 Massachusetts General Hospital, Harvard Clinical Research Institute, Boston, MA, United States of America, 8 Department of Cardiology, Washington Hospital Center, Washington, District of Columbia, United States of America, 9 Department of Cardiac Sciences, Cumming School of Medicine, Libin Cardiovascular Institute of Alberta, University of Calgary, Alberta, Canada, 10 Department of Medicine, Cardiology Division, McMaster University, PHRI-HHSC, Hamilton, Ontario, Canada

* luisecheverria@fcv.org, luisedo10@gmail.com

**Data Availability Statement:** All relevant data are within the paper and its Supporting Information files.

## Abstract

### Background

Chronic Chagas Cardiomyopathy (CCM) is a unique form of cardiomyopathy compared to other etiologies of heart failure. In CCM, risk prediction based on biomarkers has not been well-studied. We assessed the prognostic value of a biomarker panel to predict a composite outcome (CO), including the need for heart transplantation, use of left ventricular assist devices, and mortality.

### Methods

Prospective cohort study of 100 adults with different stages of CCM. Serum concentrations of amino-terminal pro-B type natriuretic peptide (NT-proBNP), galectin-3 (Gal-3), neutrophil gelatinase-associated lipocalin (NGAL), high sensitivity troponin T (hs-cTnT), soluble (sST2), and cystatin-C (Cys-c) were measured. Survival analyses were performed using Cox proportional hazard models.

### Results

During a median follow-up of 52 months, the mortality rate was 20%, while the CO was observed in 25% of the patients. Four biomarkers (NT-proBNP, hs-cTnT, sST2, and Cys-C) were associated with the CO; concentrations of NT-proBNP and hs-cTnT were associated with the highest AUC (85.1 and 85.8, respectively). Combining these two biomarkers above

**Funding:** LEE and LZR were supported by the Colombian government through Departamento Administrativo de Ciencia, Tecnología e Innovación-COLCIENCIAS (project code: 501453730398, CT 380–2011); URL: https://minciencias.gov.co/. LEE was supported by the Universität St. Gallen through the Seed Money Grants from the Leading House for Latin America of this institution (project code: 39-703).The funder had no role in study design, data collection and analysis, decision to publish, or preparation of the manuscript.

**Competing interests:** Dr. Januzzi is a Trustee of the American College of Cardiology, has received grant support from Novartis Pharmaceuticals, Roche Diagnostics, Abbott, Singulex and Prevencio, consulting income from Abbott, Janssen, Novartis, Pfizer, Merck, and Roche Diagnostics, ownership in Imbria Pharmaceuticals, and participates in clinical endpoint committees/data safety monitoring boards for Abbott, AbbVie, Amgen, Boehringer- Ingelheim, Janssen, and Takeda. This does not alter our adherence to PLOS ONE policies on sharing data and materials. The other authors have no conflict of interests to declare.

**Abbreviations:** AUC-ROC, Area under the receiver operating characteristic curve; CO, Composite Outcome; CCM, Chronic Chagas Cardiomyopathy; CD, Chagas Disease; Cys-c, Cystatin-C; ECG, Electrocardiogram; Echo, Echocardiogram; Gal-3, Galectin-3; Hs-cTnT, High sensitivity cardiac troponin T; NGAL, Neutrophil gelatinase-associated lipocalin; NT-proBNP, amino-terminal pro-B type natriuretic peptide; sST2, Soluble ST2.

their selected cut-off values significantly increased risk for the CO (HR 3.18; 95%CI 1.31–7.79). No events were reported in the patients in whom the two biomarkers were under the cut-off values, and when both levels were above cut-off values, the CO was observed in 60.71%.

## Conclusion

The combination of NT-proBNP and hs-TnT above their selected cut-off values is associated with a 3-fold increase in the risk of the composite outcome among CCM patients. The use of cardiac biomarkers may improve prognostic evaluation of patients with CCM.

## Background

Chagas Disease (CD), or American trypanosomiasis, is an infectious disease caused by the protozoan parasite Trypanosoma cruzi and currently affects approximately eight million people in endemic countries in Central and South America [1]. Moreover, an estimated 300,000 infected individuals live in the United States of America, and almost 100,000 live in the European Union with this disease. Furthermore, the pooled prevalence of infection in Latin American migrants is estimated to be around 4.2% [2–5]. Underdiagnoses of CD are frequent, estimated at up to 95% in certain areas, highlighting the need for health professionals with experience in this area [6]. The acute phase of this disease often goes under-recognized, as it is rarely severe; nevertheless, 30% of infected individuals will develop a symptomatic chronic phase characterized by severe digestive or cardiac forms of the disease [7]. Chronic Chagas cardiomyopathy (CCM) is the most common form of chronic disease; characterized by an extensive arrhythmogenic and thrombogenic status, myocardial fibrosis, segmental wall motion abnormalities, and ultimately a dilated cardiomyopathy with rapidly progressive heart failure, all of which confer high morbidity and mortality [8].

Once a patient with CD develops cardiomyopathy, predicting risk for complications (such as death or need for advanced support strategies including mechanical circulatory support or cardiac transplantation) may be challenging. A relatively under-explored option for risk prediction in CD cardiomyopathy is the use of biomarkers. Population-based studies have identified various biomarkers that may predict multiple outcomes in different pathogen-related diseases [9]. However, the experience regarding biomarkers use in CD has shown that numerous challenges still remain to allow optimal use and reliably estimate the risk of CCM progression [10, 11]. Moreover, while several serum biomarkers have been identified as having a significant diagnostic and prognostic value in heart failure (HF) of other etiologies, few studies have addressed their role in CCM, which, given its unique pathogenesis, warrants direct confirmation [12–14].

Previous work has suggested that multimodal biomarker assessment can be an important tool to improve the assessment of the prognosis in this population [15, 16]. However, data are scarce regarding the use of this strategy in patients with CD. In this context, we sought to evaluate the discriminative ability of a broad range of cardiac and renal biomarkers to assess the risk of mortality and other relevant clinical outcomes in CCM patients.

## Methods

### Study population

This prospective cohort study was performed between July 2015 to January 2020 conducted in the Heart Failure and Heart Transplant Clinic of Fundación Cardiovascular, in Floridablanca,

Colombia. The research protocol of the study was approved by the Institutional Committee on Research Ethics of the Fundación Cardiovascular de Colombia. Written consent was obtained for all patients. Adult outpatients (> 18 years old) with a positive serological diagnosis of *T. cruzi* infection (positive IgG antibodies) and echocardiographic (echo) or electrocardiographic (ECG) abnormalities consistent with chronic Chagas cardiomyopathy (left anterior fascicular block, right bundle branch block, atrioventricular blocks, ventricular premature beats, atrial fibrillation or flutter, bradycardia ≤50 h/min or echocardiographic finding suggesting myocardial alterations as evaluated by a cardiologist) were included. We enrolled patients across all the severity stages, including also individuals with implantable devices and refractory heart failure. The study sample was obtained from the CCM patients attending their follow-up evaluations; the first 100 individuals who fulfilled the inclusion criteria were enrolled. We excluded individuals with diabetes mellitus, coronary heart disease history, mitral stenosis, or uncontrolled hypertension. The Institutional Committee on Research Ethics approved the research protocol of the study. All patients provided written informed consent for their participation in the study.

## Data collection

Information regarding socioeconomic status, lifestyle factors, and medication use was recorded. Body-mass index, left ventricular ejection fraction (LVEF) calculated by Simpson's rule from four-chambers view, global longitudinal strain by speckle tracking (GLS), and estimated glomerular filtration rate (eGFR) were also measured. Finally, fasting serum samples were collected from each individual for the assessment of the six biomarkers. High sensitivity troponin T (hs-cTnT) was quantified with a 5th generation assay on an automated platform (ECLIA Elecsys 2010 analyzer, Roche Diagnostics, Germany). Galectin-3 (Gal-3) was assessed by using a quantitative method, specifically an ELFA (enzyme-linked fluorescent assay) technique (VIDAS, Biomerieux, Marcy l'Étoile, France). Amino-terminal pro-B type natriuretic peptide (NT-proBNP) levels were measured using the electrochemiluminescence method (Roche Diagnostics GmbH, Mannheim, Germany). The Alere Triage® NGAL test was used to assess Neutrophil Gelatinase-Associated Lipocalin (NGAL). Soluble ST2 (sST2) was measured from banked serum by Critical Diagnostics Presage™ sST2 assay kit via enzyme-linked immunosorbant assay (ELISA). Finally, Cystatin c (Cys-c) was quantified by an immunologic turbid metric assay (Tina-quant Cystatin C cobas®).

## Study outcomes and follow-up

The primary outcome of this study was a composite endpoint of cardiovascular mortality, heart transplant, and left ventricular assistance device (LVAD) implantation, while the secondary outcome was cardiovascular mortality. After the baseline information collection, a telephone follow-up and review of the clinical records of each patient was performed according to a standardized protocol. Investigators contacted the patients once a month during the first six months after the initial evaluation; after that, patients were called every six months. During the telephone interview, a standardized checklist of questions aimed to identify the mentioned outcomes was used.

## Statistical analysis

Categorical variables were presented as numbers and proportions, while continuous variables were reported as medians and interquartile ranges. Survival analyses were performed using the Kaplan-Meier method, life table, and Cox proportional hazard models. We considered the time to event for the CO as the number of days from enrollment to the study until the first of

the components of this outcome was reached. To identify the variables that were independently predictive of mortality, univariate and multivariate analyses using Cox's proportional regression model were performed. Due to the sample size, only age and left ventricle ejection fraction (LVEF) were included in the model, hazard ratios with its 95% confidence interval were calculated. We quantified the discriminatory ability of the biomarkers using the Harrell's C statistic and the area under the receiver operating characteristic curve (AUC-ROC). The Youden index was used to identify the best cut-off level for each biomarker(16). A p-value <0.05 was considered statistically significant for all tests. All analyses were performed using Statistical Package STATA version 15 (Station College, Texas USA).

## Results

### Population characteristics

One hundred individuals were included (55% males with a median age of 62 years [IQR 53–70]. 25% of the patients had a normal LVEF, but only 10% of the included patients had a normal global longitudinal strain (GLS) value. As expected, most of the biomarker measurements were elevated. Median NT-proBNP was 704 (IQR 170–2846) pg/mL, hs-cTnT: 11.7 (IQR 5.6–22.6) ng/L, sST2: 24.7 (IQR 20.1–31.9) ng/mL, Gal-3 had a median value of 14.2 (IQR 11.5–18.3) ng/mL, while median Cys-C and NGAL values were 1.1 (IQR 0.9–1.4) mg/L and 96.5 (IQR 69.0–145.5) ng/mL, respectively. Table 1 shows a summary of the included population characteristics by the CO.

### Incidence and rate of outcomes

After a mean of 52 months (Q1 = 42; Q3 = 54) of follow-up, there were 25 events (20 deaths, four transplants, and one LVAD implantation). The mortality in this cohort was 20% (95% CI 13.2% - 29.2%), with a mortality rate of 0.15 per 1000 person-years (95% CI 0.09–0.23). The overall incidence of this composite outcome was 25% (95% CI 16.8–34.6%), with a rate of 0.18 per 1000 person-years (95% CI 0.12–0.28). We must highlight that we did not have any loss to follow-up during this period.

### Biomarkers as predictors of adverse outcomes in CCM

All biomarkers were significantly associated with the CO (Fig 1); nevertheless, in age and LVEF adjusted models, log-transformed continuous concentrations of each biomarker were associated with the CO, except for NGAL and Galectin-3 (Table 2). Considering each biomarker dichotomously, only two evaluated biomarkers (ST2 and hs-cTnT) were significantly associated with the composite outcome (Table 2).

To analyze the potential additive value of combining sST2 and NT-proBNP measurements for predicting the composite outcome, the sample was divided into four groups based on these biomarkers cut-off points. There were only two (3.92%) deaths, HT or LVAD implantations in the patients with levels of both sST2 below 35 ng/mL and NT-proBNP under 1000 pg/ml; in contrast, the composite outcome was present in 81.25% (n = 13) of those with both these two biomarkers over the selected cut-off values, conferring a significantly higher risk when compared to those patients without sST2 or NT-proBNP elevations (HR 11.84; 95% CI 1.97–70.85) (Fig 2). In addition, the AUC for this combination was significantly higher than the ones for dichotomic sST2 (p = 0.001) and NT-proBNP (p = 0.038). Furthermore, considering that sST2 is not readily available in most clinical contexts, we analyzed the value of combined NT-proBNP and hs-TnT analysis for predicting the composite outcome (Fig 3). The results were similar to the ST2 and NT-proBNP combination, with no cases of the CO in the group of

**Table 1. Baseline characteristics the evaluated chronic Chagas cardiomyopathy patients (n = 100).**

| Variable | Composite Outcome | | p-value |
|---|---|---|---|
| | No (%) | Yes (%) | |
| **Sex** | | | |
| Female | 34 (45.33) | 11 (44) | 0.908 |
| Male | 41 (54.67) | 14 (56) | |
| **Age** (years) | 61 (53–67) | 66 (56–74) | 0.112 |
| **Area of Residence** | | | |
| Urban | 56 (74.67) | 20 (80) | 0.589 |
| Rural | 19 (25.33) | 5 (20) | |
| **BMI (kg/m$^2$)** | 26.4 (23.4–28.6) | 22.8 (19.6–28.6) | **<0.001** |
| **NYHA** | | | **0.004** |
| I | 27 (36) | 5 (20) | |
| II | 36 (48) | 7 (28) | |
| III | 11 (14.67) | 12 (48) | |
| IV | 1 (1.33) | 1 (4) | |
| **LVEF (%)** | 49 (37–60) | 25 (18–30) | **<0.001** |
| **GLS (%)** | -14.5 (-19.5; -9.3) | -7.2 (-11.1; -2.9) | **<0.001** |
| **Pharmacotherapy** | | | |
| ACEI or ARB | 51 (68) | 20 (80) | 0.252 |
| Beta-blockers | 53 (70.67) | 24 (96) | **0.009** |
| Aldosterone Antagonists | 38 (50.67) | 20 (80) | **0.010** |
| Diuretics | 28 (37.33) | 22 (88) | **<0.001** |
| Amiodarone | 21 (28.57) | 6 (25) | **0.771** |
| Digitalis | 8 (10.67) | 9 (36) | **0.003** |
| Ivabradine | 1 (1.33) | 2 (8) | 0.091 |
| Antiplatelet agents | 26 (34.67) | 3 (12) | **0.031** |
| Anticoagulants | 23 (30.67) | 14 (56) | **0.023** |
| **Biomarkers** | | | |
| NT-proBNP (pg/mL) | 352 (89–1423) | 5583 (1566–8703) | **<0.001** |
| hs-cTnT (ng/mL) | 9.2 (4.04–15.97) | 30 (17.00–69.32) | **<0.001** |
| NGAL (ng/mL) | 83 (64–112) | 147 (108–254) | **<0.001** |
| Cys-C (mg/L) | 1.05 (0.87–1.26) | 1.54 (1.22–1.89) | **<0.001** |
| Creatinine (mg/dL) | 1.08 (0.92–1.24) | 1.2 (1.03–1.54) | **0.046** |
| Galectin-3 (ng/mL) | 13.7 (10.8–16.2) | 19.6 (14.6–24.1) | **<0.001** |
| (s)ST2 (ng/mL) | 22.9 (18.7–28.6) | 37.4 (26.7–59.9) | **<0.001** |
| GFR (ml/min/1.73m$^2$) | 61 (55–61) | 56 (43–61) | **0.037** |

This table contains % for categorical variables and median (first and third quartile) for continuous variables.
Abbreviations: BMI: Body Mass Index; NYHA: New York Heart Association; LVEF: Left Ventricle Ejection Fraction;
GLS: Global Longitudinal Strain; ACEI: Angiotensin-Converting Enzyme Inhibitor; ARB: Angiotensin Receptor
Blocker; NTproBNP: N-terminal brain natriuretic propeptide; Gal-3: Galectin-3 (Gal-3); NGAL: Neutrophil
gelatinase-associated lipocalin (NGAL); sST2: Soluble ST2; Cys-c: Cystatin-C; Hs-cTnT: High sensitivity cardiac
troponin T.

patients with NT-proBNP <1000 pg/ml and Hs-TnT <15 ng/ml, while in the group of indi-
viduals with NT-proBNP >1000 pg/ml and Hs-TnT >15 ng/ml almost 61% (n = 17) had one
of the events of the CO (Fig 3). Furthermore, this group's CO risk compared to those without
hs-TnT or NT-proBNP alterations over the cut-offs was significantly higher (HR 3.18; 95%CI
1.31–7.79). The AUC value for the combined NT-proBNP and Hs-cTnT was also significantly

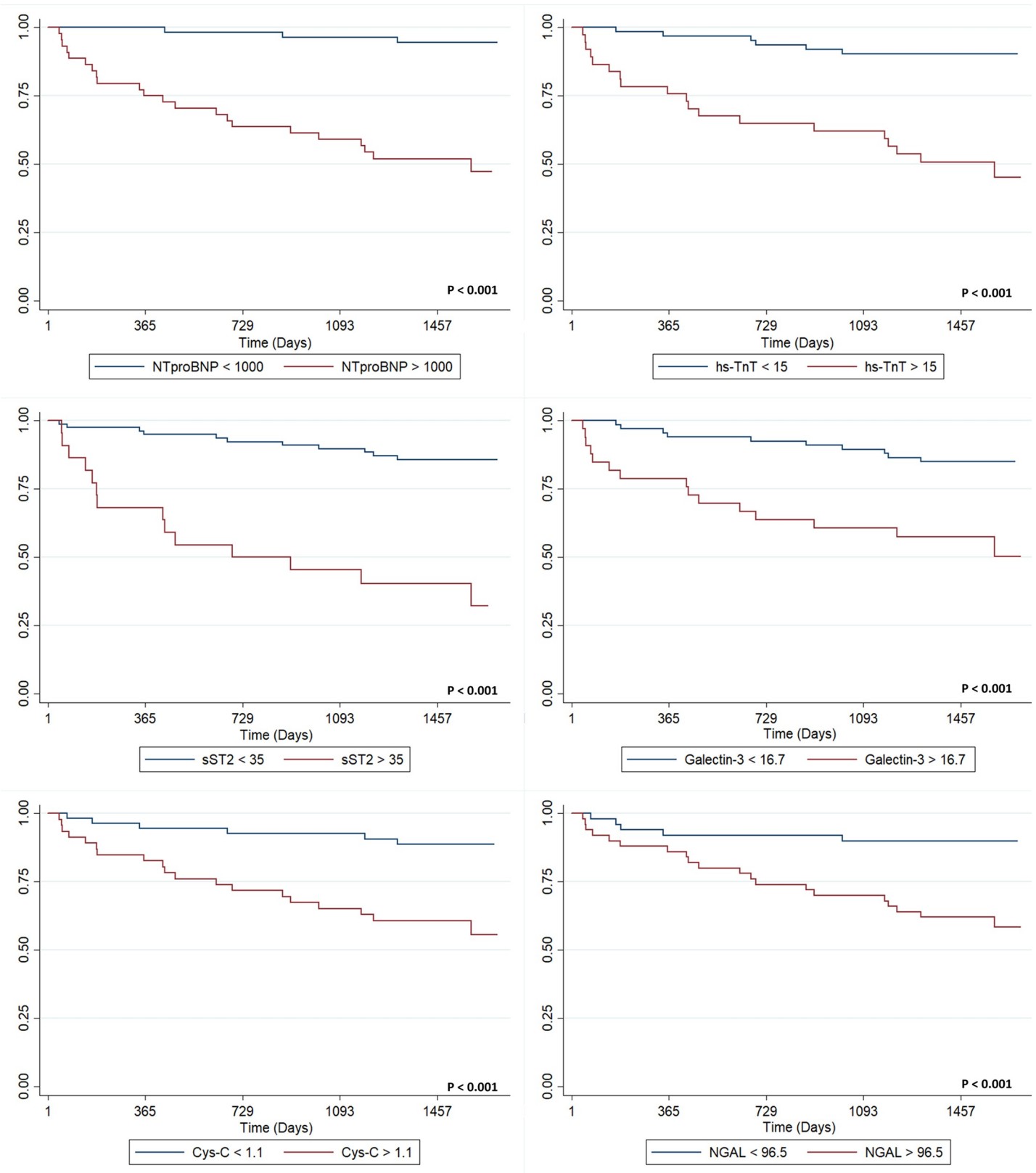

**Fig 1. Kaplan Meier survival analyses for the composite outcome stratified by biomarker results above and below selected cut-off values.** Survival by results of (A) N-terminal brain natriuretic peptide (NTproBNP); (B) High sensitivity troponin T (Hs-cTnT); (C) sST2; (D) Galectin-3; (E) Cystatin C (Cys-C); (F) Neutrophil Gelatinase-Associated Lipocalin (NGAL).

**Table 2. Prognostic value of the log-transformed biomarker levels in a continuous manner and using cut-off points for the composite outcome in patients with CCM (n = 100).**

| Biomarker | HR* | Adjusted Model | | AUC |
|---|---|---|---|---|
| | | 95% CI | p-value | |
| NT-proBNP | 2.02 | 1.43–2.85 | **0.000** | 85.06 |
| hs-cTnT | 2.65 | 1.71–4.14 | **0.000** | 85.82 |
| sST2 | 3.98 | 1.82–8.66 | **0.000** | 84.31 |
| Galectin-3 | 2.41 | 0.91–6.32 | 0.075 | 83.55 |
| Cys-C | 8.17 | 1.51–44.29 | **0.015** | 83.46 |
| NGAL | 2.05 | 0.84–4.97 | 0.112 | 82.38 |
| sST2 (>35 vs. ≤35) | 3.62 | 1.49–8.78 | **0.004** | 84.68 |
| hs-cTnT (>15 vs. ≤15) | 3.24 | 1.26–8.33 | **0.014** | 83.51 |
| Galectin-3 (>16.7 vs. ≤16.7) | 1.65 | 0.68–4.04 | 0.265 | 82.70 |
| NT-proBNP (>1000 vs. ≤1000) | 3.57 | 0.93–13.68 | 0.063 | 82.66 |
| Cys-C (>1.1 vs. ≤1.1) | 1.48 | 0.52–4.18 | 0.451 | 81.29 |
| NGAL (>96.5 vs. ≤96.5) | 1.47 | 0.51–4.211 | 0.464 | 81.39 |
| sST2 >35 + NT-proBNP >1000¥ | 11.84 | 1.97–70.85 | **0.007** | 86.10 |
| sST2 >35 + hs-cTnT >15¥ | 4.01 | 1.66–9.66 | **0.002** | 85.20 |
| NT-proBNP >1000 + hs-cTnT >15¥ | 3.18 | 1.31–7.79 | **0.011** | 84.07 |
| sST2 >35 + NT-proBNP >1000 + hs-cTnT >15¥ | 11.25 | 1.94–65.14 | **0.007** | 85.75 |

*HR adjusted by age and left ventricular ejection fraction.

¥The HR is derived from comparing the group of patients with the levels of the biomarkers over the cut-off values vs. those with the evaluated biomarkers below the selected cut-offs.

higher than the one of the dichotomic NT-proBNP (p = 0.037) but was not different from the one of the dichotomic Hs-cTnT (p = 0.064). Similar results were observed when performing a sensitivity analysis focusing on mortality as the outcome (S1 Table). Finally, the combined analysis using the three biomarkers over their selected cut-off values revealed an 11-fold increased risk of the CO (HR 11.25; 95% CI 1.94–65.14) (Fig 4).

## Discussion

This prospective cohort study represents the first study to extensively analyze the prognostic value of a series of cardiorenal biomarkers in CCM. Five serum biomarkers (NT-proBNP, hs-cTnT, sST2, Gal-3, and Cys-C) were independently associated with mortality and the CO. Additionally, NT-proBNP and hs-TnT showed the highest prognostic value in predicting the CO in this population. Of note, Hs-cTnT had a similar predictive performance to sST2; however, hs-cTnT has the advantage of being widely available in the clinical setting. A multimarker strategy has been used in other scenarios of HF patients, showing to improve the prognostic performance of every single biomarker assessment. We analyzed the value of the biomarkers combination in CCM patients, finding that the combination of NT-proBNP and sST2 or hs-cTnT significantly increased the prognostic accuracy in predicting our CO.

The need for reliable serum biomarkers for predicting outcomes such as mortality and therapeutic response in CCM has been assessed in multiple studies [9, 10, 17, 18]. Natriuretic peptides are the most studied biomarker in HF associated with CCM [12]. However, few studies have addressed the prognostic value of this and several other biomarkers in CCM. *Moreira et al.* evaluated natriuretic peptides as mortality predictors at three years of follow-up in CCM patients, finding that higher atrial natriuretic peptide (ANP) and BNP concentrations were significantly associated with a higher risk of death/heart transplant [17]. *Sherbuk et al.* confirmed

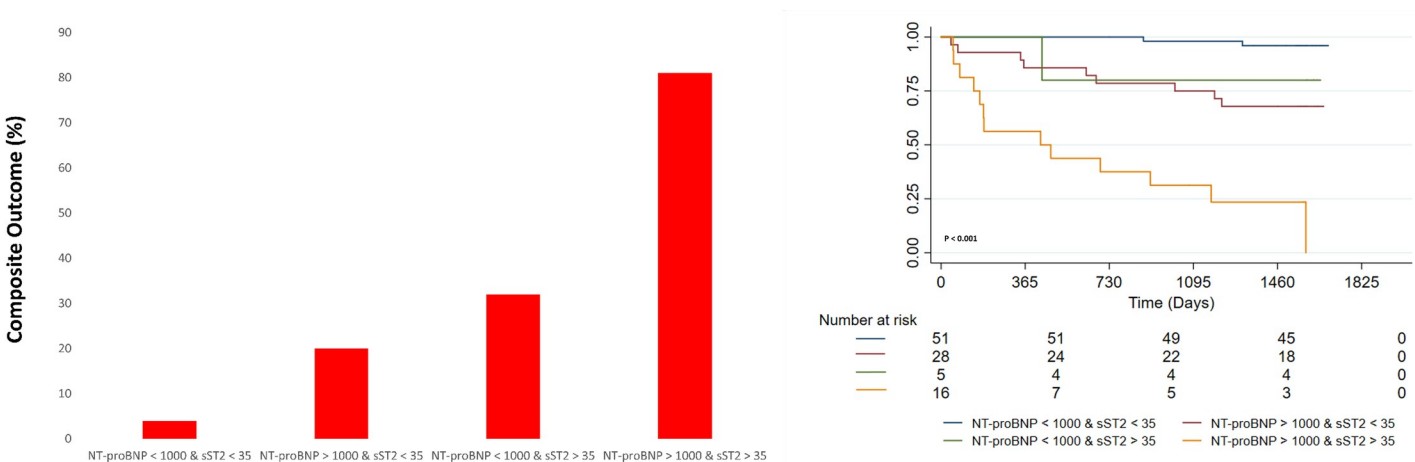

**Fig 2. Bar graph and Kaplan Meier survival analyses for the composite outcome stratified by combined NT-proBNP and ST2 according to selected cut-off values.**

these results in a study that analyzed the prognostic value of BNP, NT-proBNP, creatine kinase-myocardial band (CK-MB), troponin I, matrix metalloproteinase (MMP-2), MMP-9, and tissue inhibitor of metalloproteinases (TIMP) 1 and 2 in a group of 50 *T. cruzi*-infected Stage D Bolivian patients. In this study, higher baseline levels of BNP, NT-proBNP, CK-MB, and MMP-2 were significantly associated with increased mortality at 14 months [14]. Interestingly, in our study dichotomic Hs-cTnT had a similar prognostic value when compared to NT-proBNP and Hs-cTnT combination, a finding that may improve the assessment of CCM patients in low-resource settings, in which Hs-cTnT may be cheaper and more easily performed.

To date, no study evaluating the prognostic value of hs-cTnT, Gal-3, NGAL, sST2, and Cys-c in CCM has been reported, and our group recently reported the only study that has assessed the role of these biomarkers in CD in a cross-sectional fashion [19]. However, these serum biomarkers have already been studied in HF of other etiologies, showing promising results [20–22]. In the present study, we found that higher concentrations of these biomarkers were associated with higher risk features, and in the case of all but NGAL and Galectin-3, they were all associated with a higher risk for future cardiovascular events even after adjusting for age and

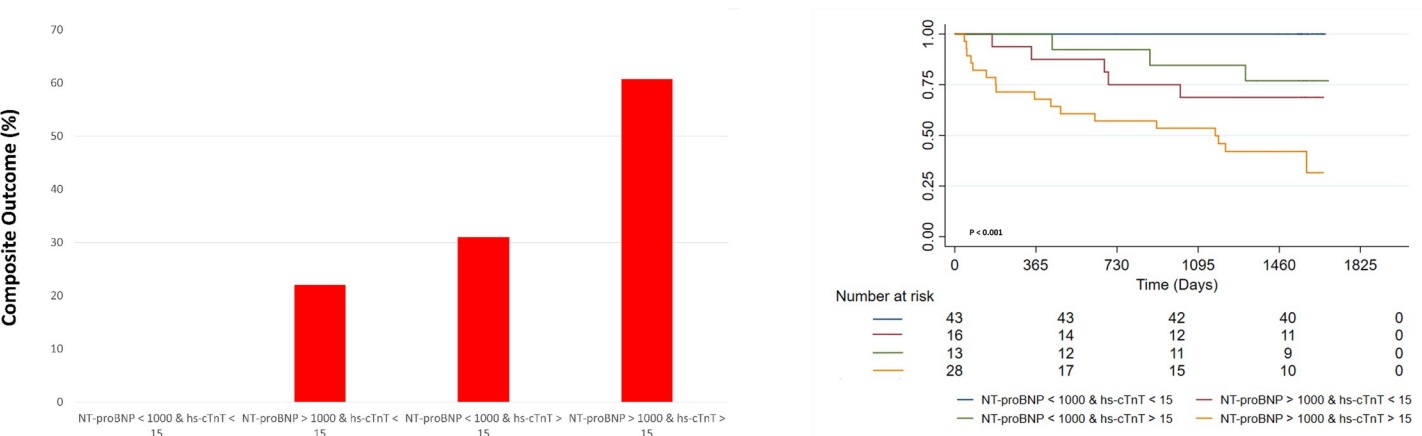

**Fig 3. Bar graph and Kaplan Meier survival analyses for the composite outcome stratified by combined NT-proBNP and Hs-cTnT according to selected cut-off values.**

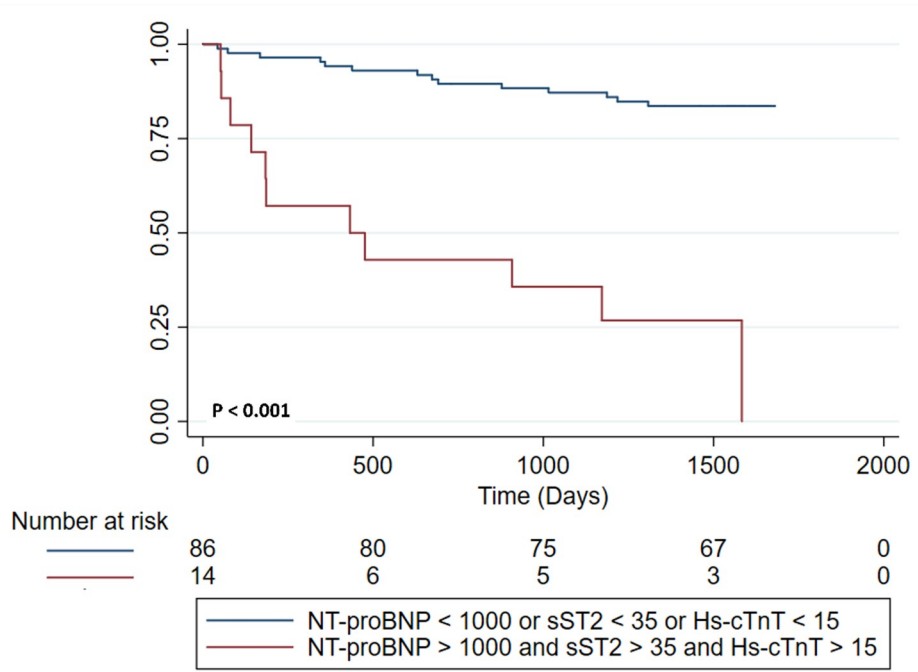

**Fig 4. Kaplan Meier survival analysis for the composite outcome stratified by combined NT-proBNP, Hs-cTnT and sST2 according to selected cut-off values.**

LVEF. Considered dichotomously in the manner a clinician might analyze biomarker results, the biomarkers studied predicted a 3 to 11-fold increase in the risk of having any composite outcome events. Given that CCM most often occurs in resource-constrained health care environments, these results might help clinicians in decision making, including consideration for transplantation evaluation.

## Study limitations

Our study has some significant limitations, including the limited sample size and the lack of inclusion of patients in the disease's indeterminate form (positive for T. cruzi, with normal ECG and echo), which could have provided insight into the role of cardiovascular biomarkers in asymptomatic individuals. In this cohort, we report a mortality rate of 0.15 per 1000 person-years at two years, which is significantly lower compared to previously reported event rates in similar studies [15]. The reasons for this difference are not apparent but may be related to differences in the compliance with optimal pharmacological treatment schemes across populations, implantable cardioverter defibrillator insertion rates, and follow-up by a specialized heart failure clinic [15, 23, 24]. Furthermore, we could not assess the role of implantable cardioverter-defibrillator use in the results, as ICD shocks may have been a relevant part of the CO considering the arrhythmogenic nature of CCM. Finally, the lack of repeated measures of the assessed biomarkers precluded assessing potential variations of the patients' clinical status during the follow-up period, for example, after changes in medical therapy or lifestyle interventions.

## Conclusion

Chronic Chagas cardiomyopathy manifests with high short-term morbidity, mortality, and significant expenditure for the healthcare system. In our prospective cohort study, we found

that four serum biomarkers (NT-proBNP, hs-cTnT, sST2, and Cys-C) were significantly associated with a higher risk of the CO among patients with CCM. The combination of NT-proBNP and sST2 or hs-cTnT provided the highest discrimination for major cardiovascular outcomes primarily driven by death. Larger multi-center studies should validate these results to identify certain biomarkers optimal for improving the prediction of adverse outcomes in CCM and potentially developing a risk score that may improve targeted therapies in this high-risk population.

## Supporting information

**S1 Graphical abstract. Central illustration or graphical abstract.** A multimarker approach combining NT-proBNP and sST2 or hs-cTnT predicted mortality and adverse cardiovascular outcomes accurately after a median follow-up of 52 months in Chronic Chagas Cardiomyopathy. The presence of two of these biomarkers over their cut-off values reflect a higher risk of mortality, indicating the need for a closer follow-up and consideration of advanced therapies. On the other hand, patients with two of these biomarkers under their cut-off values are at lower risk of adverse outcomes, potentially allowing usual follow-up and the maintenance of the guided medical therapy (GMT).
(TIF)

**S1 Table. Prognostic value of the log-transformed biomarker levels in a continuous manner and using cut-off points for the mortality outcome in patients with CCM (n = 100).**
(DOCX)

## Acknowledgments

The authors thank patients with Chagas Disease for their participation in this study.

## Author Contributions

**Conceptualization:** Luis E. Echeverría, Lyda Z. Rojas, Sergio Alejandro Gómez-Ochoa, Taulant Muka, James L. Januzzi, Rachel Marcus, Carlos A. Morillo.

**Data curation:** Lyda Z. Rojas, Sergio Alejandro Gómez-Ochoa, Oscar L. Rueda-Ochoa, Cristian David Sosa-Vesga.

**Formal analysis:** Lyda Z. Rojas, Sergio Alejandro Gómez-Ochoa, Oscar L. Rueda-Ochoa, Taulant Muka.

**Funding acquisition:** Luis E. Echeverría, Lyda Z. Rojas, Carlos A. Morillo.

**Investigation:** Luis E. Echeverría, Sergio Alejandro Gómez-Ochoa, Oscar L. Rueda-Ochoa, Taulant Muka, James L. Januzzi, Rachel Marcus, Carlos A. Morillo.

**Methodology:** Lyda Z. Rojas, Sergio Alejandro Gómez-Ochoa, Oscar L. Rueda-Ochoa, Taulant Muka, Rachel Marcus.

**Project administration:** Luis E. Echeverría, Lyda Z. Rojas, Sergio Alejandro Gómez-Ochoa.

**Resources:** Cristian David Sosa-Vesga.

**Supervision:** Luis E. Echeverría, Oscar L. Rueda-Ochoa, James L. Januzzi, Rachel Marcus, Carlos A. Morillo.

**Validation:** Luis E. Echeverría, Sergio Alejandro Gómez-Ochoa, Cristian David Sosa-Vesga, Taulant Muka, James L. Januzzi, Carlos A. Morillo.

**Visualization:** Sergio Alejandro Gómez-Ochoa.

**Writing – original draft:** Sergio Alejandro Gómez-Ochoa, Oscar L. Rueda-Ochoa, Cristian David Sosa-Vesga, James L. Januzzi, Rachel Marcus.

**Writing – review & editing:** Luis E. Echeverría, Lyda Z. Rojas, Sergio Alejandro Gómez-Ochoa, Oscar L. Rueda-Ochoa, Cristian David Sosa-Vesga, Taulant Muka, James L. Januzzi, Rachel Marcus, Carlos A. Morillo.

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
