## [Decision Letter · Decision Letter 0]

26 Jan 2021

PONE-D-20-36713

Cardiovascular Biomarkers as Predictors of Adverse Outcomes in Chronic Chagas Cardiomyopathy

PLOS ONE

Dear Dr. Echeverría,

Thank you for submitting your manuscript to PLOS ONE. After careful consideration, we feel that it has merit but does not fully meet PLOS ONE’s publication criteria as it currently stands. Therefore, we invite you to submit a revised version of the manuscript that addresses the points raised during the review process.

The Editor and the Reviewers have acknowledged the quality of your manuscript and we may consider to publish it once you provide a point-by-point response to all the issues raised.

We look forward to receiving your revised manuscript.

Kind regards,

Giuseppe Vergaro, M.D., PhD

Academic Editor

PLOS ONE

2.PLOS requires an ORCID iD for the corresponding author in Editorial Manager on papers submitted after December 6th, 2016. Please ensure that you have an ORCID iD and that it is validated in Editorial Manager. To do this, go to ‘Update my Information’ (in the upper left-hand corner of the main menu), and click on the Fetch/Validate link next to the ORCID field. This will take you to the ORCID site and allow you to create a new iD or authenticate a pre-existing iD in Editorial Manager. Please see the following video for instructions on linking an ORCID iD to your Editorial Manager account: https://www.youtube.com/watch?v=_xcclfuvtxQ

3. Please include a caption for figure 4.

5.Thank you for stating the following in the Competing Interests section:

"I have read the journal's policy and the authors of this manuscript have the following competing interests: Dr. Januzzi is a Trustee of the American College of Cardiology, has received grant support from Novartis Pharmaceuticals, Roche Diagnostics, Abbott, Singulex and Prevencio, consulting income from Abbott, Janssen, Novartis, Pfizer, Merck, and Roche Diagnostics, ownership in Imbria Pharmaceuticals, and participates in clinical endpoint committees/data safety monitoring boards for Abbott, AbbVie, Amgen, Boehringer-Ingelheim, Janssen, and Takeda. The other authors have no conflict of interests to declare."

 <h1> </h1>

Reviewers' comments:

Reviewer's Responses to Questions

**Comments to the Author**

1. Is the manuscript technically sound, and do the data support the conclusions?

Reviewer #1: Partly

Reviewer #2: Yes

2. Has the statistical analysis been performed appropriately and rigorously? 

Reviewer #1: No

Reviewer #2: No

3. Have the authors made all data underlying the findings in their manuscript fully available?

Reviewer #1: Yes

Reviewer #2: Yes

4. Is the manuscript presented in an intelligible fashion and written in standard English?

Reviewer #1: Yes

Reviewer #2: Yes

5. Review Comments to the Author

Reviewer #1: The article entitled “Cardiovascular Biomarkers as Predictors of Adverse Outcomes in Chronic Chagas Cardiomyopathy” intends to establish some biomarkers related to outcomes in chronic Chagas cardiomyopathy. Some aspects need to be addressed by the authors to clarify their results and conclusions.

1. Since Chagas Disease was abbreviated as CD, it should be included in the abbreviations list.

2. The study population selection was described as consecutive CCM patients filling the inclusion criteria, but no mention was made to those patients with pacemakers or ICD. Were those exclusion criteria?

3. It is important to define what was the clinical status of the patients during screening and inclusion. Were they only stable outpatients only or included hospitalized ones?

4. Since no other samples were obtained during follow-up and their values may be affected by medication adjustments such as increasing diuretics or beta blockers it is important to know how those adjustments may have impacted the results. A comment is needed.

5. Why hospitalizations were not part of the composite outcome? Maybe this could be a censoring point since IV medications may be used and new NYHA class achieved. Please comment on that.

6. It seems that no patient had an ICD implanted during follow-up. Since it is a high risk sample it may have occurred and could change survival. This could be treated as a censored endopoint. Please explain.

7. Finally, did you have any lost of follow-up? No censoring was noted in the tables. Please include a comment.

8. In the Methods section, the authors do not report the collection of GLS presented in the results section and also do not report the method used for left ventricular ejection fraction (unidimensional, Simpson, Teicholz, etc). The description is relevant.

9. In the follow-up procedures, you only mentioned a telephone interview for data collection. How did you confirmed a cardiovascular cause of mortality? I suggest including the structured questionnaire as supplemental material.

10. Table 1:

a. Please include in the legend that data is presented as median and interquartile range.

b. No patient received Amiodarone? This is strange in a high risk sample of CCM since arrhythmia is a frequent finding. Can you provide an explanation?

c. The group with events seems to include those with acute decompensated heart failure (NT-proBNP median value was 5583pg/mL) at inclusion. It seems that at baseline you had patients acutely ill. Can you comment on that?

d. Patients with this condition may have kidney markers Cystain C and NGAL increased due to renal involvement in an acute heart failure patient even before serum creatinine increases.

11. You must present data of univariate and multivariate analysis since it is important to known if performing those expensive tests provides an advantage beyond LVEF or GLS, traditional prognostic markers.

12. You stated a mortality of 20% in your sample and included a confidence interval that I do not understand. The same applies to your composite outcome (25%). Is this correct?

13. A potential collinearity may exist between the biomarkers studied and LVEF. Have you evaluated this?

14. The number of patients in the Kaplan-Meier curves for each interval are needed in the figures.

Reviewer #2: In this manuscript Echeverria et al explored the roles of cardiovascular biomarker as predictors of adverse outcomes in patients with Chagas Cardiomyopathy.

The study is clear and well conducted, though I have some observations:

- A clearer definition of the population included should be performed, specifying whether those patients are chronic outpatients or acute on chronic inpatients.

- Spline models should be considered to determine the optimal cut-offs for the biomarkers identified. The higher cutoffs used (i.e. 15 pg/mL for Troponin T and 1000 pg/mL for NT-proBNP) could be responsible for the non-significant HR in the dichotomized analysis.

- The number of events should also be clarified

- The adjusted model for LVEF and age is not sufficient. Other models including sex and/or GLS and especially optimal medical therapy should also be included in the model. GLS might in fact be more informative than LVEF on the prognosis. Similarly, OMT could represent a bias when interpreting survival analysis in different patients with different baseline medications.

- The three-biomarker association as predictor of outcome has only been assessed with the Cox analysis. Therefore, a Kaplan-Meier analysis of this model should also be performed.

- The Authors suggest in the Discussion section that the dichotomic troponin value has a similar prognostic value than the combined model with NT-proBNP and troponin T. However, since the two biomarkers explore two different pathophysiological backgrounds and might carry different information, their prognostic value might be affected by the underlying stage of the disease. Please, clarify.

- The quality of the figures should be increased and figure 1 with the algorithm should either made easier to read or included as a supplemental figure.

6. PLOS authors have the option to publish the peer review history of their article (what does this mean?). If published, this will include your full peer review and any attached files.

Reviewer #1: No

---

## [Author Response · Author response to Decision Letter 0]

29 Mar 2021

Response to Reviewers

Reviewer #1: The article entitled “Cardiovascular Biomarkers as Predictors of Adverse Outcomes in Chronic Chagas Cardiomyopathy” intends to establish some biomarkers related to outcomes in chronic Chagas cardiomyopathy. Some aspects need to be addressed by the authors to clarify their results and conclusions.

1. Since Chagas Disease was abbreviated as CD, it should be included in the abbreviations list.

Response: We agree with the reviewer, this abbreviation was now added to the list.

2. The study population selection was described as consecutive CCM patients filling the inclusion criteria, but no mention was made to those patients with pacemakers or ICD. Were those exclusion criteria?

Response: We appreciate this clarification, we indeed included patients with CCM and implantable devices. We have now stated this in the methods section, specifically in the “Study population” paragraph.

3. It is important to define what was the clinical status of the patients during screening and inclusion. Were they only stable outpatients only or included hospitalized ones?

Response: We thank the reviewer for this question. We have now clarified in the methods section that only outpatients were included.

4. Since no other samples were obtained during follow-up and their values may be affected by medication adjustments such as increasing diuretics or beta blockers it is important to know how those adjustments may have impacted the results. A comment is needed.

Response: We agree with the reviewer regarding this critical aspect of our study. This is a clear limitation for all studies with single measurement of biomarkers. However, all the patients included are being treated in a single center by the same group of cardiologists of the heart failure service, so we expect that changes in medical therapy may be homogeneous. We have included a small paragraph stating this in the limitations section.

5. Why hospitalizations were not part of the composite outcome? Maybe this could be a censoring point since IV medications may be used and new NYHA class achieved. Please comment on that.

Response: We understand the reviewer concerns, as hospitalizations are frequently used as part of composite outcomes in heart failure studies. However, our aim was to assess the prognostic value of the biomarkers for predicting mortality and outcomes that are tightly related to it (Heart transplant and LVAD implantation are usually last resource measures in patients without response to conventional therapy whose survival is otherwise extremely low in the short term). Considering the relatively high incidence of our composite outcome, we believe we could focus our analysis in the predictors of these outcomes, as they have a higher impact than hospitalizations.

6. It seems that no patient had an ICD implanted during follow-up. Since it is a high risk sample it may have occurred and could change survival. This could be treated as a censored endopoint. Please explain.

Response: We thank the reviewer for this question. We did not record ICD implantation as an outcome of the study, as we focused on the specific outcomes of mortality, HT and LVAD implantation. 

7. Finally, did you have any lost of follow-up? No censoring was noted in the tables. Please include a comment.

Response: We agree with the reviewer that this is a relevant aspect. As the sample size was not very large and all the patients were being treated in our center, we were able of performing a complete follow-up without losses. We have added a comment on this in the results section.

8. In the Methods section, the authors do not report the collection of GLS presented in the results section and also do not report the method used for left ventricular ejection fraction (unidimensional, Simpson, Teicholz, etc). The description is relevant.

Response: We thank the reviewer for this relevant suggestion. We have now added this information in the paragraph of “Data collection” of the Methods section.

9. In the follow-up procedures, you only mentioned a telephone interview for data collection. How did you confirmed a cardiovascular cause of mortality? I suggest including the structured questionnaire as supplemental material.

Response: We understand the reviewer´s concern. We want to clarify that we verified the cause of mortality by asking the relatives of the patient to send us the clinical records of the hospitalization in which the patient died. As most of the patients are treated in our clinic, we had direct access to their records in the institution to confirm the cause of mortality. 

10. Table 1:

a. Please include in the legend that data is presented as median and interquartile range.

Response: It is now included in the legend as follows: “This table contains % for categorical variables and median (first and third quartile) for continuous variables.”

b. No patient received Amiodarone? This is strange in a high risk sample of CCM since arrhythmia is a frequent finding. Can you provide an explanation?

Response: We agree with the reviewer this information is relevant for reporting. Therefore we have now added the data on amiodarone in the table.

c. The group with events seems to include those with acute decompensated heart failure (NT-proBNP median value was 5583pg/mL) at inclusion. It seems that at baseline you had patients acutely ill. Can you comment on that?

Response: We understand the reviewer’s concern. It is relevant to highlight that despite having elevated NT-proBNP values, patients can present with a progressive deterioration of their clinical status, not requiring hospitalization at the assessment time. For example, some of our patients come to the clinic for their outpatient assessment with recent results with NT-proBNP values of even 10000 pg/mL, and they exhibit only moderate efforts dyspnea. Therefore, although we know these are the patients with the higher risk of hospitalizations in the short term, they cannot be considered acutely ill at the moment of enrollment in the study even if they showed these high values of NT-proBNP. 

d. Patients with this condition may have kidney markers Cystain C and NGAL increased due to renal involvement in an acute heart failure patient even before serum creatinine increases.

Response: We agree with the reviewer’s comment. As Cystatin-C represents a more sensitive marker of filtration than creatinine, we expected this result. Moreover, as NGAL is a marker of tubular injury, we also expected to observe elevated values in cardiorenal syndrome patients, although it could not be amended as the only explanation for this observation.

11. You must present data of univariate and multivariate analysis since it is important to known if performing those expensive tests provides an advantage beyond LVEF or GLS, traditional prognostic markers.

Response: We thank the reviewer for this suggestion. We want to clarify that univariate and multivariate analyses are presented in Tables 1 and 2, respectively.

12. You stated a mortality of 20% in your sample and included a confidence interval that I do not understand. The same applies to your composite outcome (25%). Is this correct?

Response: We thank the reviewer for this important clarification. We observed that the confidence interval for this proportion was repeated for the two outcomes. We have now changed the intervals adequately.

13. A potential collinearity may exist between the biomarkers studied and LVEF. Have you evaluated this?

Response: We agree that this is a relevant aspect to assess. We evaluated collinearity in all the models we generated, and no evidence of it was observed between the assessed variables.

14. The number of patients in the Kaplan-Meier curves for each interval are needed in the figures.

Response: We have now included this information in the figures.

Reviewer #2: In this manuscript Echeverria et al explored the roles of cardiovascular biomarker as predictors of adverse outcomes in patients with Chagas Cardiomyopathy.

The study is clear and well conducted, though I have some observations:

1. A clearer definition of the population included should be performed, specifying whether those patients are chronic outpatients or acute on chronic inpatients.

Response: We thank the reviewer for this question. We have now clarified in the methods section that only chronic outpatients were included.

2. Spline models should be considered to determine the optimal cut-offs for the biomarkers identified. The higher cutoffs used (i.e. 15 pg/mL for Troponin T and 1000 pg/mL for NT-proBNP) could be responsible for the non-significant HR in the dichotomized analysis.

Response: Thank you very much for the suggestions to use spline models. We have used Youden index to select the cut-off points, now this was specified in statistical analysis.

3. The number of events should also be clarified

Response: Thanks for the comment now the events number was clarified in “incidence and rate of outcomes”. 

4. The adjusted model for LVEF and age is not sufficient. Other models including sex and/or GLS and especially optimal medical therapy should also be included in the model. GLS might in fact be more informative than LVEF on the prognosis. Similarly, OMT could represent a bias when interpreting survival analysis in different patients with different baseline medications.

Response: We thank the reviewer for this important comment, but given the sample size and number of events, the recommendations in the literature to include variables in a multivariate model is 10 events per variable, then the maximum that we could include in the model are two variables, otherwise we could be overestimating the model. In relation to treatment, all the patients included are being treated in a single center by the same group of cardiologists of the heart failure service, so we expect that changes in medical therapy may be homogeneous. We have included a small paragraph stating this in the limitations section.

5. The three-biomarker association as predictor of outcome has only been assessed with the Cox analysis. Therefore, a Kaplan-Meier analysis of this model should also be performed.

Response: We thank the reviewer for this suggestion. A new Kaplan Meier figure was included.

6. The Authors suggest in the Discussion section that the dichotomic troponin value has a similar prognostic value than the combined model with NT-proBNP and troponin T. However, since the two biomarkers explore two different pathophysiological backgrounds and might carry different information, their prognostic value might be affected by the underlying stage of the disease. Please, clarify.

Response: We thank the reviewer for this suggestion. Both NT-proBNP and Hs-cTnT are recommended for assessing prognosis in heart failure patients. In this context, troponin T is related to Chagas cardiomyopathy specific inflammation, which is expected to remain somehow stable across the chronic disease course. On the other hand, NT-proBNP is released as a response to myocardial tissue stretching; thus, it may be more useful for predicting outcomes in patients with dilated cardiomyopathy (Stages C and D). Therefore, we could hypothesize that NT-proBNP could have a better performance in patients with more severe forms of disease. Nevertheless, our sample size was not sufficient to adequately validate this hypothesis. We expect to prove it in future studies.

7. The quality of the figures should be increased and figure 1 with the algorithm should either made easier to read or included as a supplemental figure.

Response: We have now improved the quality of the graphics (increased the dpi) and clarified the graphical abstract legend.

---

## [Decision Letter · Decision Letter 1]

12 May 2021

PONE-D-20-36713R1

Cardiovascular Biomarkers as Predictors of Adverse Outcomes in Chronic Chagas Cardiomyopathy

PLOS ONE

Dear Dr. Echeverría,

Thank you for submitting your manuscript to PLOS ONE. After careful consideration, we feel that it has merit but does not fully meet PLOS ONE’s publication criteria as it currently stands, as some points still deserve clarification after the first review. Therefore, we invite you to submit a revised version of the manuscript that addresses the points raised during the review process.

We look forward to receiving your revised manuscript.

Kind regards,

Giuseppe Vergaro, M.D.

Academic Editor

PLOS ONE

Journal Requirements:

Reviewers' comments:

Reviewer's Responses to Questions

**Comments to the Author**

1. If the authors have adequately addressed your comments raised in a previous round of review and you feel that this manuscript is now acceptable for publication, you may indicate that here to bypass the “Comments to the Author” section, enter your conflict of interest statement in the “Confidential to Editor” section, and submit your "Accept" recommendation.

Reviewer #1: (No Response)

Reviewer #2: All comments have been addressed

2. Is the manuscript technically sound, and do the data support the conclusions?

Reviewer #1: Partly

Reviewer #2: Yes

3. Has the statistical analysis been performed appropriately and rigorously? 

Reviewer #1: No

Reviewer #2: Yes

4. Have the authors made all data underlying the findings in their manuscript fully available?

Reviewer #1: No

Reviewer #2: No

5. Is the manuscript presented in an intelligible fashion and written in standard English?

Reviewer #1: Yes

Reviewer #2: Yes

6. Review Comments to the Author

Reviewer #1: We thank the authors for addressing all questions raised by this reviewer. Some points stiil need further clarification.

1. Now you stated the refractory heart failure patients were included. Most of them in the group with events. This may explain the high number of events during the first year. This may clearly impact your results and need to be addressed.

2. TAble 1 needs to have absolute numbers included with percentagens in brackets to be consistent with the notation presented : N(%) in the first line.

3. An important point that needs clarification is the option for including AGE in the multivariate analysis. It was not significantly disticnt between the two groups, but you selected it. GLS, NYHA class and beta-blockers are highly significant in univariate analysis, but they were not selected. Can you provide an explanation?

4. Use of ICD may have a significant impact on the results. You simply stated that ICD use was not recorded. Since Cardiac transplantations and use of LVAd were described, you certainly had patients with previously implanted ICDs. Sudden cardiac death is the main reason for death in CD, so it is important to clearly identify the causes of death in your sample. An ICD shock is certainly an outcome that will be missed and may affect your results.

5. Your figures still need quality improvement.

6. The results of ROC curves described in methods need to be presented since they are the first step in the selection of those cut-off point used.

Reviewer #2: The Authors have partly addressed my concerns, though the revised figures are not presented in the revised manuscript. Nonetheless, I deem the article improved and suited for publication

7. PLOS authors have the option to publish the peer review history of their article (what does this mean?). If published, this will include your full peer review and any attached files.

Reviewer #1: No

Reviewer #2: No

---

## [Author Response · Author response to Decision Letter 1]

20 Jul 2021

Reviewer 1:

1. Now you stated the refractory heart failure patients were included. Most of them in the group with events. This may explain the high number of events during the first year. This may clearly impact your results and need to be addressed.

Response: We understand the reviewer's concerns regarding the high number of events during the first year. Beyond the inclusion of Stage D patients, we must highlight the more severe course of Chronic Chagas Cardiomyopathy compared to other heart failure etiologies. Therefore, a 25% composite outcome incidence in the evaluated period does not necessarily represent an unusual observation in this context. Furthermore, the biomarkers also identified a relevant group of patients with a very low incidence of adverse outcomes. We have now highlighted this relevant aspect in the limitations section.

2. TAble 1 needs to have absolute numbers included with percentagens in brackets to be consistent with the notation presented: N(%) in the first line.

Response: We agree with the reviewer. This has been corrected in the table.

3. An important point that needs clarification is the option for including AGE in the multivariate analysis. It was not significantly disticnt between the two groups, but you selected it. GLS, NYHA class and beta-blockers are highly significant in univariate analysis, but they were not selected. Can you provide an explanation?

Response: We thank the reviewer for this relevant recommendation. We understand his concern, so we evaluated the additive value of including the NYHA class and beta-blockers by comparing the ROC curves of the models. We observed that adding these two variables did not increase the model's ROC area for any of the biomarkers evaluated. Then, we considered that the most parsimonious model, including only age and LVEF, could be optimal due to the small sample size.

4. Use of ICD may have a significant impact on the results. You simply stated that ICD use was not recorded. Since Cardiac transplantations and use of LVAd were described, you certainly had patients with previously implanted ICDs. Sudden cardiac death is the main reason for death in CD, so it is important to clearly identify the causes of death in your sample. An ICD shock is certainly an outcome that will be missed and may affect your results.

Response: We agree with the reviewer on this aspect. However, we were unable to register information regarding ICD shocks. We have now included this in the limitations section.

5. Your figures still need quality improvement.

Response: We agree with the reviewer. This issue has been addressed in the current version. Please download the images directly from the PDF document or the submission system to allow a full-quality view.

6. The results of ROC curves described in methods need to be presented since they are the first step in the selection of those cut-off point used.

Response: We thank the reviewer for his relevant comment. The results of the AUC-ROC for each continuous biomarker assessed can be found in the last column of Table 2.

---

## [Editor Report · Decision Letter 2]

4 Oct 2021

Cardiovascular Biomarkers as Predictors of Adverse Outcomes in Chronic Chagas Cardiomyopathy

PONE-D-20-36713R2

Dear Dr. Echeverría,

We’re pleased to inform you that your manuscript has been judged scientifically suitable for publication and will be formally accepted for publication once it meets all outstanding technical requirements.

Kind regards,

Giuseppe Vergaro, M.D.

Academic Editor

PLOS ONE

---

## [Editor Report · Acceptance letter]

14 Oct 2021

PONE-D-20-36713R2 

Cardiovascular Biomarkers as Predictors of Adverse Outcomes in Chronic Chagas Cardiomyopathy 

Dear Dr. Echeverría:

I'm pleased to inform you that your manuscript has been deemed suitable for publication in PLOS ONE. Congratulations! Your manuscript is now with our production department. 

Kind regards, 

on behalf of

Dr. Giuseppe Vergaro 

Academic Editor

PLOS ONE